# Adverse birth outcomes and associated factors among newborns delivered in Sao Tome & Principe: A case–control study

**Alexandra Vasconcelos**[1]*, **Swasilanne Sousa**[2], **Nelson Bandeira**[3], **Marta Alves**[4], **Ana Luísa Papoila**[4], **Filomena Pereira**[1], **Maria Céu Machado**[5]

1 Instituto de Higiene e Medicina Tropical (IHMT), Unidade de Clínica Tropical—Global Health and Tropical Medicine (GHTM), Universidade NOVA de Lisboa, Lisboa, Portugal, 2 Department of Pediatrics, Hospital Dr. Ayres de Menezes, São Tomé, República Democrática de São Tomé e Príncipe, 3 Department of Obstetrics & Gynecology, Hospital Dr. Ayres de Menezes, São Tomé, República Democrática de São Tomé e Príncipe, 4 NOVA Medical School/Faculdade de Ciências Médicas, CEAUL, Universidade NOVA de Lisboa, Lisboa, Portugal, 5 Faculdade de Medicina de Lisboa, Universidade de Lisboa, Lisboa, Portugal

* alexandravasc@gmail.com

## Abstract

### Background

Newborns with one-or-more adverse birth outcomes (ABOs) are at greater risk of mortality or long-term morbidity with health impacts into adulthood. Hence, identifying ABO-associated factors is crucial for devising relevant interventions. For this study, ABOs were defined as prematurity (PTB) for gestational age <37 weeks, low birth weight (LBW) <2.5 kg, macrosomia >4 kg, asphyxia for a 5-minute Apgar score <7, congenital anomalies, and neonatal sepsis. This study aimed to assess factors associated with ABOs among babies delivered at the only hospital of Sao Tome & Principe (STP), a resource-constrained sub-Saharan-Central African country.

### Methods

A hospital-based unmatched case–control study was conducted among newborns from randomly selected mothers. Newborns with one-or-more ABO were the cases (ABO group), while healthy newborns were the controls (no-ABO group). Data were collected by a face-to-face interview and abstracted from antenatal care (ANC) pregnancy cards and medical records. Multivariable logistic regression analysis was performed to identify ABO-associated factors considering a level of significance of α = 0.05.

### Results

A total of 519 newborns (176 with ABO and 343 no-ABO) were enrolled. The mean gestational age and birthweight of cases and controls were 36 (SD = 3.7) weeks with 2659 (SD = 881.44) g and 39.6 (SD = 1.0) weeks with 3256 (SD = 345.83) g, respectively. In the multivariable analysis, twin pregnancy [aOR 4.92, 95% CI 2.25–10.74], prolonged rupture of membranes [aOR 3.43, 95% CI 1.69–6.95], and meconium- fluid [aOR 1.59, 95% CI 0.97–

**Data Availability Statement:** All relevant data are within the paper and its Supporting Information files.

**Funding:** AV was supported by the Fundação para a Ciência e Tecnologia (FCT) (https://www.fct.pt/index.phtml.pt/), grant number SFRH/BD/117037/2016. The funder had no role in the study design, data collection and analysis, decision to publish or preparation of the manuscript.

**Competing interests:** The authors have declared that no competing interests exist.

2.62] were significantly associated with ABOs. Eight or more ANC contacts were found to be protective [aOR 0.33, 95% CI 0.18–0.60, p<0.001].

## Conclusion

Modifiable factors were associated with ABOs in this study and should be considered in cost-effective interventions. The provision of high-quality ANC should be a priority. Twin pregnancies and intrapartum factors such as prolonged rupture of membranes and meconium-stained amniotic fluid are red flags for ABOs that should receive prompt intervention and follow-up.

## Introduction

Adverse birth outcomes (ABOs) are a major global public health problem linked to child mortality and morbidity since they can impact children's short- and long-term well-being due to neurological and health problems throughout their life course [1,2]. Inevitably, a newborn with an ABO is at a higher risk for mortality than newborns without an ABO [3]. Additionally, ABOs may disrupt the family condition, leading to high individual and social costs [4].

The magnitude of ABOs worldwide has dramatically decreased in recent decades, although a large gap still exists between high-income and low- and middle-income countries (LMICs), making birth outcomes important measures of health [4]. The specific burden of each ABO can vary according to country specificities, although preterm birth (PTB) is the most well-accepted benchmark for morbidity attributable to early gestation [5]. In recent years, ABOs in LMICs have received attention, with a wide range of ABOs being reported across different studies [6–11]. While some studies include indicators for early gestation, such as PTB, fetal growth restriction, low birth weight (LBW) as well as perinatal mortality and fetal loss/miscarriage [4,5], others exclusively analyze live newborns at birth [12]. Furthermore, ABOs can coexist and share the same underlying risk factors, are mostly multifactorial, physiologically diverse, and not entirely well understood, despite decades of research [5].

Different studies have revealed that diverse risk factors are associated with ABOs [1,4,5,8]. Studies in LMICs have reported numerous sociodemographic factors, maternal characteristics, previous pregnancy outcomes, neonatal factors, and socioeconomic and health system-related factors [8,12,13]. For example, mothers with previous pregnancy outcomes of PTB or LBW are more likely to have recurrence of these ABOs than those without previous PTB or LBW babies [6,9]. A study on maternal health during pregnancy found that women who had at least one comorbidity during their pregnancy had a twofold higher risk of delivering LBW babies than women without any health problems [14]. Indeed, studies report that antepartum infections such as malaria, syphilis and others, and noninfectious conditions such as anemia, hypertension, hyperglycemia, and obstetric complications are all linked to ABOs [9–12]. Additionally, lack of adequate antenatal care (ANC), household air pollution from unclean cooking fuels, open defecation, no access to improved water, violence, and other socioeconomic disparities are also considered important risk factors for ABOs in sub-Saharan African (SSA) countries [15–18].

Sao Tome & Principe (STP) is an SSA country with limited data on the overall ABO rate at the country level, and in the current era of the Sustainable Development Goals (SDGs), neonatal mortality remains high, demanding urgent intervention in ABO reduction [19–21]. As discussed above, most risk factors contributing to ABOs are amenable to modification, although they are not the same across different cultures and socioeconomic statuses within a society

[4,18]. Thus, knowing STP context-specific reality of ABOs enables to target and implement the most proper evidence-based interventions for this setting. For this study, ABOs were defined as PTB, LBW, macrosomia, major congenital anomaly, birth asphyxia and neonatal sepsis suspicion. STP has significant health resource-constraints, since there are no blood cultures or other microbiologic techniques available, and other procedures such as umbilical arterial blood gas analysis, are not feasible [22,23]. Moreover, there are no neonatal intensive care units and, similar to other units in LMICs, lacks a mechanical ventilator or any continuous positive airway pressure machine and surfactant therapy for neonatal care [24,25]. Therefore, birth asphyxia is only determined by using the APGAR score [26–29], and early-onset neonatal sepsis diagnosis is done in a suspicion-based algorithm [24,25].

Thus, this study aimed to improve neonatal health outcomes in this very resource-constrained SSA country by identifying the factors associated with ABOs among newborns delivered at the only hospital maternity unit in STP. Thus, knowledge of the burden of ABOs and key associated factors in this specific setting can be used to design targeted interventions and better allocate resources for effective ABO reduction.

## Materials and methods

### Study design and period

A facility-based unmatched case–control study was conducted in STP among 519 newborns whose mothers gave birth at Hospital Dr. Ayres de Menezes (HAM) Maternity Unit. The recruitment of newborns' mothers occurred from July 2016 to November 2018.

### Setting

The archipelago of STP, the smallest Central SSA country, is a 219 161-inhabitant country, with a young population and an annual birth cohort of approximately 6.521 babies [19]. The rate of deliveries in health units is approximately 98%, with 82.4% occurring at the HAM maternity unit, the only hospital in the country [20]. The HAM is a tertiary healthcare facility and receives complicated cases referred from facilities with lower levels of care, as it is the only facility with Comprehensive Emergency Obstetric Care (CEmOC) capable of providing blood transfusions and performing cesarean sections.

The maternity unit has a resource-constrained unit for small and sick newborn babies— Newborn Care Unit (NCU)—with six baby cots, in which babies receive oxygen through nasal prongs or face masks and are assisted by general doctors and nurses, since there are no neonatologists and only two pediatricians in STP.

### Participants

The eligibility criteria for participants were as follows: 1) all neonates delivered at HAM and 2) newborns who were born outside the hospital but were later admitted at HAM on the day of birth. A total of 535 newborns were initially enrolled.

The exclusion criteria included the following: 1) all neonates delivered at HAM born without any signs of life (stillbirths), 2) newborns whose mothers had cognitive impairment, and 3) adolescent or illiterate mothers who had not obtained permission from their parents or legal guardians to participate in the study. Sixteen met the exclusion criteria (stillbirths), with a total of 519 participants enrolled.

## Selection of cases and controls

Cases (ABO group) were newborns with at least one adverse birth outcome: PTB, LBW, macrosomia, birth asphyxia, congenital anomaly and/or probable neonatal sepsis (see Operational definition of variables). Controls (no-ABO group) were healthy newborns without adverse birth outcomes ($\geq$ 37 gestational weeks at birth, weight $\geq$ 2.5 kg at birth and not greater than 4 kg, 5-min APGAR score $\geq$ 7, no congenital anomaly, and no probable sepsis).

## Sample size determination and sampling procedures

Sample size followed the WHO-steps approach [30] applying a web-based sample size calculator, Raosoft, which suggested a minimum sample size of S = 355, which placed the right dimension between 355 (95%) and 579 (99%) confidence [31]. A total of 535 participants were enrolled based on the following assumptions: two-sided 95% confidence level, and power of 80% to detect an odds ratio of at least 2 for ABOs. This sample size was also supported by PASS software [32].

A random sampling was applied to recruit the newborn's mother, selecting every second interval folder from the pile of mothers´ medical folders, and occurred during daytime hours of working days. The mothers who consented were interviewed after recovery from the delivery, and the mother-newborn dyads were followed-up throughout their stays until hospital discharge.

## Data collection tools, procedures, and quality control

Data were collected by a pre-tested, structured interviewer-administered questionnaire developed from the STP Demographic and Health Survey (DHS) and similar studies [18,19,20]. Additionally, a standard abstraction checklist from ANC pregnancy cards and maternal and newborn records was conducted. The questionnaire contains information related to the sociodemographic characteristics of parents, preconception healthcare services and obstetrics-related characteristics of the mothers.

To ensure data quality, pretesting was performed on 10 cases and 13 controls and modified based on the findings, mainly adjusting terminology for more culturally friendly terms. The questionnaires were checked for completeness and consistency. Data collection was regularly reviewed by the supervisors. The principal investigator (a pediatrician) executed and was responsible for the field activities as follows: i) obtaining consent and enrollment of the mothers, ii) data collection from ANC cards plus maternal and newborn records, iii) newborns' clinical exams (for diagnosis confirmation), iv) face-to-face interviews, and v) entry of all data collection into the app survey tool. Finally, double data entry was performed to minimize errors during data entry.

## Operational definition of variables

**ABOs**: 1) preterm was defined as a birth that occurred before 37 completed weeks (less than 259 days) of gestation [33]; 2) LBW as a weight of < 2.5 kg at birth [34,35]; 3) macrosomia as a birth-weight over 4000 g irrespective of gestational age [36]; 4) birth asphyxia as an APGAR score at 5-minutes inferior to seven [26,27]; 5) congenital anomaly as structural changes in one or more parts of the infant's body that are present at birth [16]; and/or 6) probable neonatal sepsis or early-onset neonatal sepsis was defined through a clinical-based algorithm as having one or more of the following: i) newborns with early suspicious signs and symptoms (hypothermia or fever, lethargy, poor perfusion, hypotonia, bulging fontanel, respiratory

distress, apnea, and gasping respiration), ii) with or without an identifiable maternal infectious risk, and/or iii) requiring NCU admission and antibiotic treatment [37,38].

**Maternal infectious risk:** operationally defined as i) maternal fever (axillary temperature >37.9 C) at the time of delivery, and/or ii) prolonged rupture of membranes (PROM) (≥18 hours) [34], and/or iii) meconium or foul-smelling amniotic fluid [39].

**Gestational age:** estimated from the date of onset of the last normal menstrual period or through ultrasound dating of pregnancy [12,40].

**Maternal anemia:** hemoglobin concentration <11 g/dl.

**Intrapartum conditions:** included fetal malpresentation [41], umbilical cord complications [42], PROM [37,43], meconium-stained amniotic fluid [44], postpartum hemorrhage (>500 mL bleeding), preeclampsia (hypertension ≥140/90 mmHg and proteinuria in women who were normotensive at ANC), and obstructed labor [45]).

## Data analysis

Data were entered into the QuickTapSurvey app (2010–2021 Formstack), an offline survey app tool, and exported to Excel to be further checked for completeness, coded, entered, and cleaned. Data were analyzed using SPSS version 25. In this study, cases were coded as 1, and controls were coded as 0 for analysis. The proportion of missing data ranged from 0.8 to 10% across variables. Missing values higher than 10% were described in the analysis.

Descriptive statistics were used to describe the frequency distribution of each of the variables mentioned earlier. The chi-square test was used to compare the proportion of cases and controls between selected categorical variables. To identify factors associated with ABO, univariable and multivariable logistic regression analyses were performed. In the univariable analysis, explanatory variables with a p value <0.25 were candidates for a multivariable logistic regression model to monitor the influence of confounding variables. With their 95% confidence intervals (95% CI), crude (cOR) and adjusted (aOR) odds ratios were determined to assess the strength and existence of an association. The level of significance $\alpha = 0.05$ was considered.

## Ethics approval and consent to participate

The study complies with the Declaration of Helsinki and was approved and consented to by dedicated ethics oversight bodies such as the Ministry of Health of STP and by the main board of HAM, since at the time the study protocol was submitted, there was no ethics committee in STP. All methods in our study were performed in accordance with the relevant guidelines and regulations in practice. Written informed consent was obtained from all participants after the purpose of the research was explained orally by the researcher. Approval by the participants´ parents or legal guardians was asked in the case of adolescents under 16 years of age or for illiterate women. The participants or their legal representatives also consented to have the results of this research work published. Participation in the survey was voluntary, as participants could decline to participate at any time during the study.

## Results

A total of 519 newborns (176 cases and 343 controls) were enrolled. The newborn's mean gestational age (GA) was 38.73 weeks with a standard deviation (SD) of 2.62 (minimum 25—maximum 43 weeks). The mean birth weight was 3053.79 ±(SD = 649) g (minimum 900 g–maximum 4650 g). Cases had a mean GA and birth weight of 36.02 ±(SD = 3.7) and 2659.66 ±(SD = 881.44) g, respectively, while their counterparts had 39.61 ±(SD = 1.03) weeks and 3256.02 ±(SD = 345.83) g, respectively. The mean maternal age was 26.5 years with a standard

deviation of 7.03 (minimum 14—maximum 43 years old). The mean maternal age for cases and controls was 26.99 (SD = ±7.15) and 26.24 (SD = ±6.96), respectively.

## Adverse birth outcomes

The current study revealed that 343 (66%) births were healthy live births, while the remaining 176 (34%) were births with child-related adverse birth outcomes. Regarding ABOs (Table 1), 92 (17.7%) were PTB, 83 (16%) had LBW, and 42 (8.1%) had birth asphyxia. Eight babies (1.5%) had congenital anomalies, 21 (4%) had a probable neonatal sepsis and 21 (4%) had a birth weight higher than four kilograms (macrosomia).

The maternal characteristics as well as antepartum, intrapartum, and postpartum factors for all participants and for cases versus controls are described in Table 2.

## Factors associated with adverse birth outcomes

In the univariable logistic regression, number of ANC contacts, twin pregnancy, delivery assisted by midwives or obstetricians, preeclampsia, PROM, obstructed labor, cesarean section, meconium-stained amniotic fluid, and infectious risk were eligible for multivariable analysis (Table 3).

In the multivariable logistic analysis, twin pregnancy (aOR 4.92, 95% CI 2.25–10.74; p<0.001), PROM (aOR 3.43, 95% CI 1.69–6.95, p = 0.001), and meconium-stained amniotic fluid (aOR 1.59, 95% CI 0.97–2.62, p = 0.068) were independently significantly associated with ABOs (Table 3). Having 8 or more ANC contacts was found to be a protective factor (aOR 0.33, 95% CI 0.18–0.60, p<0.001), as shown in Table 3.

## Discussion

Assessing neonatal ABOs and identifying contributing factors can help avoid neonatal mortality and morbidity thoroughly and thoughtfully. As a result, the goal of this study was to identify the factors related to ABOs in neonates admitted at birth to HAM in the capital city of

**Table 1. Frequency and types of adverse birth outcomes related to the newborns among deliveries attended in Hospital Dr. Ayres de Menezes, Sao Tome & Principe.**

| Adverse birth outcomes | | Frequency | Percent |
|---|---|---|---|
| Preterm baby or birth | No | 427 | 82.3 |
| | Yes | 92 | 17.7 |
| Low Birth Weight | No | 436 | 84 |
| | Yes | 83 | 16 |
| Macrosomia | No | 498 | 96.0 |
| | Yes | 21 | 4.0 |
| Congenital anomaly | No | 511 | 98.5 |
| | Yes | 8 | 1.5 |
| Birth asphyxia | No | 477 | 91.9 |
| | Yes | 42 | 8.1 |
| Probable neonatal sepsis | No | 498 | 95.9 |
| | Yes | 21 | 4.0 |
| ABOs observed in a new birth | no ABO | 343 | 66 |
| | at least 1 ABO | 176 | 34 |
| | Total | 519 | 100.0 |

Abbreviations: ABOs—adverse birth outcomes.

**Table 2. Maternal characteristics, antepartum, intrapartum, and postpartum factors for all participants and for cases (newborns with ABO) versus controls (newborns with no-ABO).**

| Variables | TOTAL n = 519 (%) | ABO n = 176 n (%) | no-ABO n = 343 n (%) | *p*-value |
|---|---|---|---|---|
| **Sociodemographic factors** | | | | |
| Age | | | | 0.277 |
| 14–16 | 29 (5.6) | 11 (6.3) | 18 (5.2) | |
| 17–19 | 74 (14.3) | 21 (11.9) | 53 (15.5) | |
| 20–34 | 330 (63.6) | 108 (61.4) | 222 (64.7) | |
| ≥35 | 86 (16.6) | 36 (20.5) | 50 (14.6) | |
| Maternal education | | | | 0.004 |
| none | 19 (3.7) | 7 (4) | 12 (3.5) | |
| primary | 298 (57.4) | 111 (63.1) | 187 (54.5) | |
| secondary | 166 (32) | 40 (22.7) | 126 (36.7) | |
| higher | 36 (6.9) | 18 (10.2) | 18 (5.2) | |
| Maternal occupation | | | | 0.921 |
| housewife | 326 (62.8) | 111 (63.1) | 215 (62.7) | |
| student | 36 (6.9) | 11 (6.3) | 25 (7.3) | |
| employed | 157 (30.3) | 54 (30.7) | 103 (30) | |
| Marital status | | | | 0.650 |
| union/married | 410 (79) | 137 (77.8) | 273 (79.6) | |
| single | 109 (21) | 39 (22.2) | 70 (20.4) | |
| Baby´s father education | | | | 0.163 |
| none | 11 (2.1) | 3 (1.7) | 8 (2.3) | |
| primary | 178 (34.3) | 65 (36.9) | 113 (32.9) | |
| secondary | 126 (24.3) | 39 (22.2) | 87 (25.4) | |
| higher | 38 (7.3) | 19 (10.8) | 19 (5.5) | |
| unknown | 166 (32) | 50 (28.4) | 116 (33.8) | |
| Residence | | | | 1.000 |
| urban | 231 (45.3) | 77 (45.3) | 154 (45.3) | |
| rural | 279 (54.7) | 93 (54.7) | 186 (54.7) | |
| Improved water | | | | 0.917 |
| yes | 377 (72.6) | 127 (72.7) | 250 (72.9) | |
| no | 142 (27.4) | 49 (27.8) | 93 (27.1) | |
| Sanitation | | | | 0.516 |
| yes | 272 (52.4) | 96 (54.5) | 176 (51.3) | |
| open defecation | 247 (47.6) | 80 (45.5) | 167 (48.7) | |
| **Preconception factors** | | | | |
| Pregnancy planned | | | | 0.631 |
| yes | 126 (24.3) | 41 (23.3) | 85 (24.8) | |
| no | 293 (56.5) | 97 (55.1) | 196 (57.1) | |
| missing | 100 (19.3) | 38 (21.6) | 62 (18.1) | |
| Ever use of family planning methods | | | | 0.721 |
| yes | 106 (24.6) | 33 (23.2) | 73 (25.3) | |
| no | 325 (75.4) | 109 (76.8) | 216 (74.7) | |
| Gravidity | | | | 0.389 |
| 1 | 126 (24.3) | 37 (21) | 89 (25.9) | |
| 2–5 | 273 (52.6) | 94 (53.4) | 179 (52.2) | |
| ≥5 | 120 (23.1) | 45 (25.6) | 75 (21.9) | |

(*Continued*)

**Table 2.** (*Continued*)

| Variables | TOTAL n = 519 (%) | ABO n = 176 n (%) | no-ABO n = 343 n (%) | *p*-value |
|---|---|---|---|---|
| Parity | | | | |
| 0 | 153 (29.5) | 50 (28.4) | 103 (30) | 0.622 |
| 1–4 | 316 (60.9) | 106 (60.2) | 210 (61.2) | |
| ≥5 | 50 (9.6) | 20 (11.4) | 30 (8.7) | |
| Previous abortion | | | | 0.158 |
| yes | 156 (30.1) | 60 (34.1) | 96 (28.0) | |
| no | 363 (69.9) | 116 (65.9) | 247 (72.0) | |
| Previous stillbirth | | | | 0.068 |
| yes | 53 (10.2) | 24 (13.6) | 29 (8.5) | |
| no | 466 (89.8) | 152 (86.4) | 314 (91.5) | |
| Poor birth spacing | | | | 0.815 |
| yes | 100 (19.3) | 35 (19.9) | 65 (19.0) | |
| no | 419 (80.7) | 141 (80.1) | 278 (81) | |
| Previous cesarean section | | | | 0.592 |
| yes | 15 (2.9) | 6 (3.4) | 9 (2.6) | |
| no | 504 (97.1) | 170 (96.6) | 334 (97.4) | |
| **ANC** | | | | |
| GA at first ANC visit | | | | 0.835 |
| ≤12 | 272 (62.4) | 91 (61.5) | 181 (62.8) | |
| >12 | 164 (37.6) | 57 (38.5) | 107 (37.2) | |
| Number of ANC contacts | | | | **<0.001** |
| 1–4 | 75 (14.6) | 31 (18.1) | 44 (12.9) | |
| 5–7 | 237 (46.2) | 101 (59.1) | 136 (39.8) | |
| ≥ 8 | 201 (39.2) | 39 (22.8) | 162 (47.4) | |
| Obstetric ultrasound | | | | 0.777 |
| yes | 212 (40.8) | 106 (60.2) | 201 (58.6) | |
| no | 208 (40.4) | 70 (39.8) | 142 (41.4) | |
| Twin pregnancy | | | | **<0.001** |
| yes | 34 (6.6) | 24 (13.6) | 10 (2.9) | |
| no | 485 (93.4) | 152 (86.4) | 333 (97.1) | |
| Maternal anemia | | | | 0.978 |
| yes | 161 (31) | 54 (30.7) | 107 (31.2) | |
| no | 240 (46.2) | 81 (46) | 159 (46.4) | |
| not done | 118 (22.7) | 41 (23.3) | 77 (22.4) | |
| Bacteriuria | | | | 0.212 |
| yes | 155 (29.9) | 44 (25.0) | 111 (32.4) | |
| no | 207 (39.9) | 74 (42.0) | 133 (38.8) | |
| not done | 157 (30.3) | 58 (33.0) | 99 (28.9) | |
| Hyperglycemia | | | | 0.538 |
| yes | 16 (3.1) | 7 (4.0) | 9 (2.6) | |
| no | 346 (66.7) | 113 (64.2) | 233 (67.9) | |
| not done | 157 (30.3) | 56 (31.8) | 101 (29.4) | |

(*Continued*)

**Table 2.** (Continued)

| Variables | TOTAL n = 519 (%) | ABO n = 176 n (%) | no-ABO n = 343 n (%) | *p*-value |
|---|---|---|---|---|
| Malaria | | | | 1.000 |
| yes | 3 (0.6) | 1 (0.6) | 2 (0.6) | |
| no | 385 (74.2) | 131 (74.4) | 254 (74.1) | |
| not done | 131 (25.2) | 44 (25) | 87 (25.4) | |
| HIV | | | | 1.000 |
| yes | 3 (0.6) | 1 (0.6) | 2 (0.6) | |
| no | 486 (99.4) | 164 (99.4) | 322 (99.4) | |
| Syphilis | | | | 0.340 |
| yes | 5 (1.1) | 3 (1.9) | 2 (0.6) | |
| no | 463 (98.9) | 154 (98.1) | 309 (99.4) | |
| HsbAg | | | | 0.810 |
| yes | 16 (3.1) | 4 (2.3) | 12 (3.5) | |
| no | 300 (57.8) | 103 (58.5) | 197 (57.4) | |
| not done | 203 (39.1) | 69 (39.2) | 134 (39.1) | |
| **Health facility-related factors** | | | | |
| Baby delivered at HAM | | | | 0.592 |
| yes | 504 (97.1) | 170 (96.6) | 334 (97.4) | |
| no | 15 (2.9) | 6 (3.4) | 9 (2.6) | |
| Transferred from another unit | | | | **0.009** |
| yes | 21 (4.0) | 13 (7.4) | 8 (2.3) | |
| no | 498 (96) | 163 (92.6) | 335 (97.7) | |
| Delivery assisted by | | | | **0.011** |
| obstetrician | 84 (16.2) | 39 (22.8) | 45 (13.5) | |
| midwife | 421 (83.4) | 132 (77.2) | 289 (86.5) | |
| **Intrapartum complications and mode of delivery** | | | | |
| Fetal malpresentation | | | | 0.607 |
| yes | 4 (0.8) | 2 (1.1) | 2 (0.6) | |
| no | 515 (99.2) | 174 (98.9) | 341 (99.4) | |
| PROM | | | | |
| yes | 38 (7.3) | 22 (12.5) | 16 (4.7) | **0.002** |
| no | 481 (92.7) | 154 (87.5) | 327 (95.3) | |
| Pre/Eclampsia | | | | **0.029** |
| yes | 37 (7.1) | 19 (10.8) | 18 (5.2) | |
| no | 482 (92.9) | 157 (89.2) | 325 (94.8) | |
| Meconium-stained amniotic fluid | | | | **0.027** |
| yes | 90 (17.3) | 40 (22.7) | 50 (14.6) | |
| no | 429 (82.7) | 136 (77.3) | 293 (85.4) | |
| Umbilical cord complication | | | | 0.355 |
| yes | 34 (6.6) | 14 (8.0) | 20 (5.8) | |
| no | 485 (93.4) | 162 (92) | 323 (94.2) | |
| Obstructed labor | | | | 0.122 |
| yes | 52 (10) | 23 (13.1) | 29 (8.5) | |
| no | 467 (90) | 153 (86.9) | 314 (91.5) | |

(*Continued*)

**Table 2.** (Continued)

| Variables | TOTAL n = 519 (%) | ABO n = 176 n (%) | no-ABO n = 343 n (%) | p-value |
|---|---|---|---|---|
| Postpartum hemorrhage | | | | 0.339 |
| yes | 1 (0.2) | 1 (0.6) | 0 | |
| no | 518 (99.8) | 175 (99.4) | 343 (100) | |
| Normal Vaginal delivery | | | | **0.034** |
| yes | 433 (83.4) | 138 (78.4) | 295 (86.0) | |
| no | 86 (16.6) | 38 (21.6) | 48 (14.0) | |
| Cesarean section | | | | **0.039** |
| yes | 79 (15.2) | 35 (19.9) | 44 (12.8) | |
| no | 440 (84.8) | 141 (80.1) | 299 (87.2) | |
| Instrumental vaginal delivery | | | | 0.694 |
| yes | 7 (1.3) | 3 (1.7) | 4 (1.2) | |
| no | 512 (98.7) | 173 (98.3) | 339 (98.8) | |
| **Newborns characteristics and complications** | | | | |
| Gestational Age | | | | |
| <32 | 16 (3.1) | 16 (9.1) | 0 | **<0.001** |
| 32 a 37 | 76 (14.6) | 76 (43.2) | 0 | |
| 38–41 | 350 (67.4) | 65 (36.9) | 285 (83.1) | |
| ≥41 | 77 (14.8) | 19 (10.8) | 58 (16.9) | |
| Sex | | | | 0.308 |
| feminine | 253 (48.7) | 80 (45.5) | 173 (50.4) | |
| masculine | 266 (51.3) | 96 (54.5) | 170 (49.6) | |
| Birth weight | | | | **<0.001** |
| <1500 g | 17 (3.3) | 17 (9.7) | 0 | |
| 1500–2499 g | 66 (12.7) | 66 (37.5) | 0 | |
| 2500–3999 g | 415 (80) | 72 (40.9) | 343 (100) | |
| ≥ 4000 g | 21 (4.0) | 21 (11.9) | 0 | |
| IUGR | | | | **<0.001** |
| yes | 21 (4.0) | 18 (10.2) | 3 (0.9) | |
| no | 498 (96) | 158 (89.8) | 340 (99.1) | |
| Infectious risk | | | | **<0.001** |
| yes | 117 (22.5) | 62 (35.2) | 55 (16) | |
| no | 402 (77.5) | 114 (64.8) | 288 (84) | |
| Neonatal resuscitation performed | | | | **<0.001** |
| yes | 28 (5.4) | 26 (14.8) | 2 (0.6) | |
| no | 491 (94.6) | 150 (85.2) | 341 (99.4) | |
| Fetal distress at birth | | | | **<0.001** |
| yes | 103 (19.8) | 72 (40.9) | 31 (9.0) | |
| no | 416 (80.2) | 104 (59.1) | 312 (91.0) | |
| Admission at NCU | | | | **<0.001** |
| yes | 70 (13.5) | 64 (36.4) | 6 (1.7) | |
| no | 449 (86.5) | 112 (63.6) | 337 (98.3) | |

(*Continued*)

**Table 2.** (Continued)

| Variables | TOTAL n = 519 (%) | ABO n = 176 n (%) | no-ABO n = 343 n (%) | *p*-value |
|---|---|---|---|---|
| Received antibiotic | | | | **<0.001** |
| yes | 112 (21.6) | 72 (40.9) | 40 (11.7) | |
| no | 407 (78.4) | 104 (59.1) | 303 (88.3) | |

Bold text for p-value ≤0.05.

Abbreviations: GA–gestational age; ANC–antenatal care; PROM–prolonged rupture of membranes; IUGR–intrauterine growth restriction; NCU–neonatal care unit; HAM–Hospital Dr. Ayres de Menezes.

Note: Most pregnant women received tetanus toxoid vaccination, iron supplementation and blood pressure measurements, and these factors were not considered. All mothers stated no consanguinity with the baby´s father and did not have smoking habits; therefore, these factors were also not included.

STP. Multiple births (twins), meconium-stained amniotic fluid and PROM were all identified as significant associated factors for ABOs in the current study.

This study showed a rate of 17.7% PTB and 16% LBW, which are higher than the published estimates for STP of 12% PTB and 6.6% LBW [2,40,46]. In resource-constrained countries, one explanation for the difference found is the misclassification of late preterm babies (gestational age <37 weeks) in term babies [40,47]. Since gestational age in LMICs is mainly estimated from the date of onset of the last normal menstrual period, most times without ultrasound confirmation, mistakes are frequent [47].

Regarding LBW rates in the country, the source used is the UNICEF Multiple Indicator Cluster Surveys (MICS) [20,21]. MICS data rely upon the information provided by the mother about last birth in the previous two years and are associated with a higher risk of recall bias; therefore, these rate discrepancies between this study and official STP data can be linked to the above reasons. The lack of difference between the rate of PTB and LBW in this study is in line with the studies done in Nepal [48], Ethiopia [49] and Kenya [14] as biologically, a PTB has a higher risk of having LBW as they are less likely to get sufficient time for maturity, growth, and nutrient intake. Another important consideration is that most PTBs in this study were "late preterm" (76/92), often associated with a normal postnatal clinical course with no relevant complications, when compared to very preterm babies (16/92) [50].

The 1.5% congenital malformation and 4% macrosomia rates found in this study are similar to those described by other LMICs [4,8,36].

The neonatal sepsis rate published for LMICs ranges between 1.6% and 3.8% of all live births [50], with disparities among studies regarding the clinical algorithms used and the lack of gold-standard blood cultures for diagnosis [51,52]. This study 4% rate is in line with the overall LMIC rate but neonatal sepsis rates in STP are probably much higher since only babies diagnosed between the 24th and 36th hours of life were included; thus, all those with onset of sepsis after being discharged were missed out.

The 8.1% rate of birth asphyxia found in this study is lower when compared to other LMICs: a study in Ethiopia reports a pooled prevalence of 22.52%, 18% in other East African countries and 9.1% in some Central African countries [53,54]. The lower burden of asphyxia in STP may be due to differences in case definition, as in this study, birth asphyxia was only based on a fifth minute APGAR score less than 7, whereas other studies also used other criteria, such as umbilical cord pH < 7- or 20-min Apgar score less than 7 or multiorgan failure in the first 72 h or convulsion in the first 24 h of life [53]. APGAR scoring is also vulnerable to midwife evaluation and therefore susceptible to higher scoring for better health-related outcomes [26,27].

**Table 3.** Factors associated with adverse birth outcomes among newborns delivered at Hospital Dr. Ayres de Menezes in Sao Tome & Principe.

| Variables | ABO n = 176 n (%) | no-ABO n = 343 n (%) | cOR (95% CI) | p-value | aOR (95% CI) p-value |
|---|---|---|---|---|---|
| **Sociodemographic characteristics** | | | | | |
| Maternal education | | | | | |
| none | 7 (4) | 12 (3.5) | 1 | | |
| primary | 111 (63.1) | 187 (54.5) | 1.02 (0.39–2.66) | 0.972 | |
| secondary | 40 (22.7) | 126 (36.7) | 0.54 (0.20–1.48) | 0.232 | |
| higher | 18 (10.2) | 18 (5.2) | 1.71 (0.55–5.35) | 0.353 | |
| **ANC** | | | | | |
| Number of ANC contacts | | | | | |
| 1–4 | 31 (18.1) | 44 (12.9) | 1 | | |
| 5–7 | 101 (59.1) | 136 (39.8) | 1.05 (0.62–1.78) | 0.845 | 1.03 (0.59–1.78), p = 0.922 |
| ≥ 8 | 39 (22.8) | 162 (47.4) | 0.34 (0.19–0.61) | <0.001 | 0.33 (0.18–0.60), p <0.001 |
| Twin pregnancy | | | | | |
| yes | 24 (13.6) | 10 (2.9) | 5.26 (2.45–11.27) | <0.001 | 4.92 (2.25–10.74) p <0.001 |
| no | 152 (86.4) | 333 (97.1) | 1 | | |
| **Intrapartum complications** | | | | | |
| PROM | | | | | |
| yes | 22 (12.5) | 16 (4.7) | 2.92 (1.49–5.72) | 0.002 | 3.43 (1.69–6.95), p = 0.001 |
| no | 154 (87.5) | 327 (95.3) | 1 | | |
| Pre/Eclampsia | | | | | |
| yes | 19 (10.8) | 18 (5.2) | 2.18 (1.12–4.28) | 0.023 | |
| no | 157 (89.2) | 325 (94.8) | 1 | | |
| Meconium-stained amniotic fluid | | | | | |
| yes | 40 (22.7) | 50 (14.6) | 1.72 (1.08–2.74) | 0.021 | 1.59 (0.97–2.62), p = 0.068 |
| no | 136 (77.3) | 293 (85.4) | 1 | | |
| **Health facility-related factors** | | | | | |
| Transferred from another unit | | | | | |
| yes | 13 (7.4) | 8 (2.3) | 3.34 (1.36–8.22) | 0.009 | |
| no | 163 (92.6) | 335 (97.7) | 1 | | |
| Delivery assisted by | | | | | |
| obstetrician | 39 (22.8) | 45 (13.5) | 1 | | |
| midwife | 132 (77.2) | 289 (86.5) | 0.53 (0.33–0.85) | 0.008 | |
| Normal Vaginal delivery | | | | | |
| yes | 138 (78.4) | 295 (86) | 1 | | |
| no | 38 (21.6) | 48 (14.0) | 1.69 (1.06–2.71) | 0.029 | |
| Cesarean section | | | | | |
| yes | 35 (19.9) | 44 (12.8) | 1.69 (1.04–2.74) | 0.035 | |
| no | 141 (80.1) | 299 (87.2) | 1 | | |
| **Newborns complications*** | | | | | |
| IUGR | | | | | |
| yes | 18 (10.2) | 3 (0.9) | 12.91 (3.75–44.47) | <0.001 | |
| no | 158 (89.8) | 340 (99.1) | 1 | | |
| Infectious risk | | | | | |
| yes | 62 (35.2) | 55 (16.0) | 2.85 (1.87–4.35) | <0.001 | |
| no | 114 (64.8) | 288 (84.0) | 1 | | |
| Neonatal resuscitation | | | | | |
| yes | 26 (14.8) | 2 (0.6) | 29.55 (6.93–126.11) | <0.001 | |

*(Continued)*

**Table 3.** (Continued)

| Variables | ABO<br>n = 176<br>n (%) | no-ABO<br>n = 343<br>n (%) | cOR (95% CI) | p-value | aOR (95% CI) p-value |
|---|---|---|---|---|---|
| no | 150 (85.2) | 341 (99.4) | 1 | | |
| Fetal distress at birth | | | | | |
| yes | 72 (40.9) | 31 (9.0) | 6.97 (4.33–11.22) | <0.001 | |
| no | 104 (59.1) | 312 (91.0) | 1 | | |
| Admission at NCU | | | | | |
| yes | 64 (36.4) | 6 (1.7) | 32.09 (13.53–76.13) | <0.001 | |
| no | 112 (63.6) | 337 (98.3) | 1 | | |
| Received antibiotic | | | | | |
| yes | 72 (40.9) | 40 (11.7) | 5.24 (3.36–8.19) | <0.001 | |
| no | 104 (59.1) | 303 (88.3) | 1 | | |

Abbreviations: ANC–antenatal care; PROM–prolonged rupture of membranes; IUGR–intrauterine growth restriction; NCU–neonatal care unit; cOR: Crude odds ratio; aOR–adjusted odds ratio; CI–confidence interval.

* Neonatal complications were related as a consequence of the case definition and therefore were not included in the multivariable model.

The fivefold higher risk of multiple pregnancies having ABOs can be associated with the fact that monochorionic pregnancies have a vascular anastomosis within the placenta, affecting the perfusion of each twin and promoting adverse outcomes such as preterm labor, premature rupture of membranes, antepartum hemorrhage and fetal death [55]. Adverse outcomes in twin pregnancies were also reported in other studies [55–57]. As an example, some studies report that twin pregnancies have a thirteen-fold increase in rates of stillbirth in monochorionic pregnancies and a fivefold increase in dichorionic twins compared with singleton pregnancies [57]. Hence, the association between multiple pregnancies and ABO in this study reveals the importance of screening for multiple pregnancies as one key component of ANC to reduce the risk of ABO [56]. Other interventions to reduce ABOs related to twin pregnancies that can be recommended in STP are delivery at 37 weeks gestation in uncomplicated dichorionic twin pregnancies and delivery at 36 weeks in monochorionic pregnancies, as proposed by Cheong-See F et al. [57].

In this study, complete ANC with eight or more contacts was a protective factor, which should be expected, since ANC is the most important practice for mothers to obtain more information about nutrition and health, to perform screenings and to learn about danger signs regarding pregnancy and childbirth [16,22]. If a mother lacks ANC, minor obstetric conditions are not detected and managed early; therefore, serious complications and ABOs will likely develop [22,58]. Our finding is similar to the results from Ethiopia and other LMICs, in which ABOs, especially PTB and LBW, were higher among mothers with few ANC contacts [8,18,22,59]. This cutoff is linked to the fact that eight or more ANC contacts can reduce perinatal deaths by up to 8 per 1000 births when compared to only four ANC visits [60].

The rate of pregnancies with meconium-stained amniotic fluid in this study (17.3%) is similar to the international standard and from other studies in LMICs [10,61,62]. The occurrence of meconium-stained amniotic fluid during labor has long been considered a predictor of ABOs and an important sign of fetal distress associated with high rates of neonatal resuscitation, respiratory distress, lower Apgar score, birth asphyxia, neonatal care unit admissions and meconium aspiration syndrome [10,63]. Additionally, approximately 5–10% of neonates with meconium will experience meconium aspiration syndrome, which accounts for approximately 12% of neonatal mortality (as much as a 40% case fatality rate for the neonate and

approximately 2% of perinatal mortality) as well as for neonatal sepsis and pulmonary disease, representing an important risk factor not only for ABOs but also for death [10,63]. The odds of experiencing an ABO, in this study, were identified as being approximately twofold higher among meconium-stained fluid compared with a clear amniotic fluid at birth. Hence, early detection by using a latent follow-up chart and partograph and timely intervention is recommended to reduce this significant risk factor [10,63].

PROM was defined as rupture of the membrane lasting more than 18 hours before labor [37,43]. The overall rate in this study was 7.3%, which is in line with the estimated 5%-10% of all pregnancies in LMICs [64,65]. It was also identified as a significant risk factor for ABO, with a threefold higher risk compared to newborns without PROM. This is a well-known risk factor associated with early-onset neonatal sepsis and increased risk for perinatal mortality [64–67]. Thus, preventive measures should focus on the recognition of these high-risk newborns with early treatment with empirical antibiotics [64–67]. Such approaches would be a safe and cost-effective strategy, especially in STP where there are no laboratory culture techniques available.

In this study, no association was found between maternal infectious diseases (malaria, HIV, syphilis) and ABOs. This is mainly due to the fact that STP, in contrast to most SSA countries, is about to reach the elimination of HIV, syphilis, hepatitis B and malaria supporting the reduced number of mothers infected enrolled in this study [68,69]. Other viruses known to cause ABOs in SSA countries, such as Zika and dengue were not evaluated in this study since there is no evidence of Zika virus transmission in the country, and the first dengue fever outbreak in STP occurred two years after the completion of the field study [70]. Understanding the role and impact of environmental factors in ABOs is important, and contrary to expectations, no association was found between maternal sanitation behavior (open defecation) or type of access to water (no-improved) and ABOs, as reported in other LMICs studies [15,17,67].

## Strengths and limitations

In this study, all clinical pregnancy-related information was abstracted from ANC cards and maternity registers to limit recall bias. Moreover, this study analyzed the main variables that are most frequently associated with ABOs in LMICs, including factors related to i) maternal health conditions, ii) infectious diseases, iii) environmental factors, iv) healthcare access, and v) obstetric interventions.

Regarding the limitations, maternal factors such as gestational weight gain, BMI, and height were not analyzed in this study, as these measurements are not performed during ANC contacts in the country, missing key mediators for ABOs [12]. Participants were asked about the weight gained during the pregnancy or their height, but mothers were not aware of either their height or weight gain. This lack of knowledge of basic one´s features and body conscience might also be due to the lack of scales and stadiometers in most healthcare facilities.

Notwithstanding these limitations, this study represents a significant first step in the description of ABOs and key modifiable associated factors in this resource-constrained setting.

## Future research

The scope of this case-control study missed-out some recent areas of research, such as the impact of water contamination, traditional herbs, exposure to toxins on birth outcomes and long-term implications of ABOs on child development, growth, and health outcomes, which should be a matter of future research in this country [15–17]. This will promote further

understanding of the complex determinants of ABOs and provide new clues on how to improve birth outcomes and maternal and child health.

## Conclusions

The high rates of ABOs could be reduced through the provision of high-quality intrapartum care, as twin pregnancy, meconium-stained amniotic fluid and PROM were identified as factors significantly associated with ABOs in this study. Having an ANC with 8 or more contacts was found to be protective, reinforcing the key strategy to reduce ABOs by providing a complete ANC service throughout the continuum of care.

Therefore, the modifiable associated factors found in this case-control study should be considered in cost-effective interventions. Reducing this ABO burden will not only impact neonatal mortality rates but will also promote child well-being, growth, and favorable health outcomes across their life course and provide substantial population-level human capital returns in this small and resource-constrained SSA country.

## Supporting information

**S1 File.**
(XLSX)

## Acknowledgments

A special remark for the late Professor João Luís Baptista PhD MD—AV research cosupervisor —a great man who was a thinker and a fighter for Africa's improvement of public health. We are indebted to all the women who participated in the study. The authors would like to thank Elizabeth Carvalho and the 1) medical team and nurses of Hospital Dr. Ayres de Menezes Maternity for their support, especially to the chief-nurse Paulina Oliveira, and 2) Ana Sequeira, Rita Coelho, Ana Margalha, Ana Castro, Alexandra Coelho, and Inês Gomes for field support. We would like to acknowledge Instituto Camões, I.P. for the logistic support in Sao Tome & Principe.

## Author Contributions

**Conceptualization:** Alexandra Vasconcelos, Filomena Pereira, Maria Céu Machado.

**Data curation:** Marta Alves.

**Formal analysis:** Marta Alves, Ana Luísa Papoila.

**Funding acquisition:** Alexandra Vasconcelos.

**Investigation:** Alexandra Vasconcelos.

**Methodology:** Alexandra Vasconcelos, Ana Luísa Papoila, Filomena Pereira, Maria Céu Machado.

**Project administration:** Alexandra Vasconcelos.

**Resources:** Swasilanne Sousa, Nelson Bandeira.

**Software:** Marta Alves, Ana Luísa Papoila.

**Supervision:** Ana Luísa Papoila, Filomena Pereira, Maria Céu Machado.

**Validation:** Ana Luísa Papoila.

**Visualization:** Swasilanne Sousa, Nelson Bandeira.

**Writing – original draft:** Alexandra Vasconcelos.

**Writing – review & editing:** Alexandra Vasconcelos, Swasilanne Sousa, Nelson Bandeira, Marta Alves, Ana Luísa Papoila, Filomena Pereira, Maria Céu Machado.

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
