## [Decision Letter · Decision Letter 0]

30 Mar 2023

PONE-D-22-27043Adverse birth outcomes and associated factors among newborns delivered in a western African country: a case‒control studyPLOS ONE

Dear Dr. Vasconcelos,

Thank you for submitting your manuscript to PLOS ONE. After careful consideration, we feel that it has merit but does not fully meet PLOS ONE’s publication criteria as it currently stands. Therefore, we invite you to submit a revised version of the manuscript that addresses the points raised during the review process.

We look forward to receiving your revised manuscript.

Kind regards,

Calistus Wilunda, DrPH

Academic Editor

PLOS ONE

Additional Editor Comments:

Please clarify the exposure variables. Some of the variables are mediators and it is unclear why they have been included in the analysis. The model building strategy is not clear.

Some other important risk factors such as maternal nutritional status are missing.

Provide the name of the country, instead of ‘western African”

Reviewers' comments:

Reviewer's Responses to Questions

**Comments to the Author**

1. Is the manuscript technically sound, and do the data support the conclusions?

Reviewer #1: Partly

Reviewer #2: Yes

2. Has the statistical analysis been performed appropriately and rigorously? 

Reviewer #1: Yes

Reviewer #2: Yes

3. Have the authors made all data underlying the findings in their manuscript fully available?

Reviewer #1: No

Reviewer #2: No

4. Is the manuscript presented in an intelligible fashion and written in standard English?

Reviewer #1: No

Reviewer #2: No

5. Review Comments to the Author

Reviewer #1: I commend the authors for their effort on such an important study in STP. The method and findings and just. However, more work needs to be done on the style and presentation in English to meet journals expectations. There is a need to be succinct and use less repetition. Some sentences and paragraphs are too chunky. I have made individual comments on the attached word document.

Reviewer #2: 1. Specify the study country, where it applies.

2. The entire draft needs to be rewritten and/or edited for more clarity for the flow and content of respective section.

3. Reconsider the measurement of ANC as 8+ ANC is the latest development and its implementation may not be widely adopted, especially for births that happened around tail end of your data collection period.

4. Could it be possible to enhance the internal validity of the findings by incorporating additional 'distal' factors, such as SES and environmental factors, into the analysis? This could provide valuable contributions to the field as the current associations have already been established in OBGYN textbooks. Moreover, testing hypotheses on these distal factors may reveal some ‘unique and original’ insights for the field under consideration.

6. PLOS authors have the option to publish the peer review history of their article (what does this mean?). If published, this will include your full peer review and any attached files.

Reviewer #1: **Yes: **Lydia Sandrah Kaforau

Reviewer #2: **Yes: **Dr Amanuel Abajobir

---

## [Author Response · Author response to Decision Letter 0]

14 May 2023

Rebuttal letter

Point-by-point response to Reviewers

Dear Editor and Reviewers, 

On behalf of all authors, we express our gratitude to you for the constructive review of our manuscript PONE-D-22-27043

Adverse birth outcomes and associated factors among newborns delivered in a western African country: a case‒control study

PLOS ONE

We truly appreciate the Academic Editor Calistus Wilunda DrPH and Reviewer´s Lydia Sandrah Kaforau and Dr Amanuel Abajobir feedback and precious contributions.

We now submit a revised version of the manuscript that addresses the points raised during the review process.

In the following detailed response, we address each comment calling for changes point-by-point, indicating where relevant additional texts have been added to the body of the manuscript. 

We submit a manuscript with track changes (file labeled 'Revised Manuscript with Track Changes') and a clean version of the review manuscript (“Manuscript”). 

We believe that with the changes made, the manuscript has improved substantively in view of being considered for publication.

Journal Requirements:

and https://journals.plos.org/plosone/s/file?id=ba62/PLOSOne_formatting_sample_title_authors_affiliations.pdf

Authors’ response: 

Our manuscript now meets PLOS ONE's style requirements.

Additional Editor Comments:

Please clarify the exposure variables. Some of the variables are mediators and it is unclear why they have been included in the analysis. The model building strategy is not clear.

Some other important risk factors such as maternal nutritional status are missing.

Provide the name of the country, instead of ‘western African”

Authors’ response: 

Provide the name of the country, instead of ‘western African”

We amended the title as suggested: 

Adverse birth outcomes and associated factors among newborns delivered in Sao Tome & Principe: a case‒control study

Some other important risk factors such as maternal nutritional status are missing.

Some risk factors were not possible to collect since they are not included in the WHO´s ANC package for settings such as Sao Tome & Principe. Therefore, maternal factors such as gestational weight gain, BMI, and height were not feasible to include. During the face-to-face interviews I asked the participants if they Knew how many kg they might have gained during the pregnancy or their height, but participants did not know their height or weight. The main reason is that in Sao Tome & Principe, it is not frequent to find scales and stadiometers in health care facilities.

This is disclosed in the “Strengths and Limitations" section.

Please clarify the exposure variables. Some of the variables are mediators and it is unclear why they have been included in the analysis. The model building strategy is not clear.

The exposure variables (sociodemographic characteristics, preconception factors, ANC, intrapartum complications, health facility-related factors) for our case-control study were selected according to the literature, clinical practice and similar studies conducted in resource-constrained settings. For example, 

i) Adane, et al. Adverse birth outcomes among deliveries in Gonder university hospital, Northwest Ethiopia. BMC Pregnancy Childbirth. 2014;14:90. http://www.biomedcentral.com/1471-2393/14/90

ii) Hailemichael HT, Debelew GT, Alema H, et al. Determinants of adverse birth outcome in Tigrai region, North Ethiopia: Hospital-based case-control study. BMC Pediatrics. 2020;20, 10. https://doi.org/10.1186/s12887-019-1835-6

iii) KC A, Basel PL, Singh S. Low birth weight and its associated risk factors: Health facility-based case-control study. PloS one. 2020;15(6):e0234907. https://doi.org/10.1371/journal.pone.0234907

Newborns’ characteristics and complications were described as we wanted to characterize our study population for these variables. These variables may be considered potential mediators, which means they are part of the causal pathway between the exposure variable and the outcome variable. The decision to include potential mediators in the analysis was due to the type of research question for this case-control study. 

However, none of these potential mediator variables were included in the multivariable model. A note at the end of Table 3 adds to clarify this. 

By including potential mediators, researchers can better understand the underlying mechanisms or pathways that are involved in the relationship between the exposure variable and the outcome variable. This helped to identify potential targets for interventions. 

Reviewers' comments:

Reviewer's Responses to Questions

Comments to the Author

1. Is the manuscript technically sound, and do the data support the conclusions?

Reviewer #1: Partly

Reviewer #2: Yes

2. Has the statistical analysis been performed appropriately and rigorously?

Reviewer #1: Yes

Reviewer #2: Yes

3. Have the authors made all data underlying the findings in their manuscript fully available?

Reviewer #1: No

Reviewer #2: No

Authors’ response: 

The datasets used and analyzed during the current study are attached as “supporting information”.

4. Is the manuscript presented in an intelligible fashion and written in standard English?

Reviewer #1: No

Reviewer #2: No

Authors’ response: 

We now submit a revised manuscript with American style. 

Issues of grammar and wrong spelling were all rectified. 

An additional effort was made with the help of an English-speaking health professional who assisted in improving the language of the paper now submitted. 

We also submitted the manuscript to the AJE Academic Services of American Journal Experts.

We believe that with the changes made, the manuscript is now improved substantively.

5. Review Comments to the Author

Reviewer #1: 

I commend the authors for their effort on such an important study in STP. The method and findings and just. However, more work needs to be done on the style and presentation in English to meet journals expectations. There is a need to be succinct and use less repetition. Some sentences and paragraphs are too chunky. I have made individual comments on the attached word document.

Authors’ response: 

We truly appreciate Reviewer 1´s valuable contributions and feedback. 

We now submit a revised manuscript with improved language that is much more succinct and without repetition. We were also consistent with the use of terms and their abbreviations throughout the text.

We amended the manuscript according to all Reviewer 1´s suggestions.

We believe that this paper will be very important for Sao Tome & Principe and similar countries. 

Reply to Reviewer 1´s suggestions:

Abstract was amended as Reviewer 1 suggested: lines 31 and 33.

Introduction was amended as reviewer 1 suggested: line 169 from material and methods were incorporated in the introduction and were also reworded.

Materials and methods were amended as Reviewer 1 suggested: line 157, 158, 169

All material and methods sections were summarized and compressed as suggested. Repetitions were deleted. 

Results were amended as Reviewer 1 suggested: we deleted repetitions. Prevalence was deleted.

Table 2 - for ANC characteristics was unformatted. The point estimate can now be properly seen. The text presented in the Results section was summarized and repetitions were deleted. 

Discussion was amended as Reviewer 1 suggested: abbreviations were used instead of the terms. Misclassification of outcomes related to prematurity was further elaborated. Authors and references were cited accordingly. 

Regarding the comment “What you mean by authors? Do you meant to say clinicians?”

“Some authors even propose a cesarean section when there is a thick meconium-stained amniotic fluid to ensure a better outcome for the neonate even in the presence of normal fetal heart rate tracings on cardiotocography [55].” It is recommended in the following article: 55. Desai D, Maitra N, Patel P. Fetal heart rate patterns in patients with thick meconium staining of amniotic fluid and its association with perinatal outcome. Int J Reprod Contracept Obstet Gynecol. 2017;6(3):1030–5” 

We decided to delete this information since this C-section indications are not always feasible in countries such as STP that have only one anesthesiologist. 

References: All references were renumbered according to the changes in their new order of citation in the manuscript. 

Reviewer #2: 

Authors’ response: 

We also appreciate Reviewer 2´s precious contributions and feedback. 

1. Specify the study country, where it applies.

Authors’ response: 

Done. 

2. The entire draft needs to be rewritten and/or edited for more clarity for the flow and content of respective section.

Authors’ response: 

Done. We now submit a revised manuscript with American style and issues of grammar and wrong spelling were all rectified. 

An additional effort was made with the help of an English-speaking health professional who assisted in improving the language of the paper now submitted. We also submitted the manuscript to the AJE Academic Services of the American Journal Experts. We believe that with the changes made the manuscript is now improved substantively.

3. Reconsider the measurement of ANC as 8+ ANC is the latest development and its implementation may not be widely adopted, especially for births that happened around tail end of your data collection period.

Authors’ response:

We appreciate Reviewer 2´s comment. 

We analyzed our study participants by ANC measurement of [1-4], [4-7] and [8+] ANC contacts.

We used these measurements since they translate the country´s reality, as we have previously found, in another study that we conducted that: “Antenatal care coverage in Sao Tome & Principe is a success compared to other SSA countries [8, 44]. Attendances are extremely high for a low-resource country. The minimum of eight ANC contacts, endorsed by the 2016 WHO´s guideline, was reached in almost forty percent. Moreover, 85.1% of the participants had four or more contacts when the rates reported in SSA for 4 visits were approximately 62% [44].”

Vasconcelos A, Sousa S, Bandeira N, Alves M, Papoila AL, Pereira F, Machado MC. Antenatal screenings and maternal diagnosis among pregnant women in Sao Tome & Principe-Missed opportunities to improve neonatal health: A hospital-based study. PLOS Global Public Health. 2022;2. 

Available from: https://doi.org/10.1371/journal.pgph.0001444

4. Could it be possible to enhance the internal validity of the findings by incorporating additional 'distal' factors, such as SES and environmental factors, into the analysis? This could provide valuable contributions to the field as the current associations have already been established in OBGYN textbooks. Moreover, testing hypotheses on these distal factors may reveal some ‘unique and original’ insights for the field under consideration.

Authors’ response: 

We were able to include all main sociodemographic variables recognized as being associated with ABOs (maternal age, education, occupation, marital status, baby´s father´s education, residence) and some environmental factors such as access to improved water and type of sanitation. During the mothers’ face-to-face interviews, we also collected other distal factors, such as the type of cooking fuel mothers used. This has been associated with ABOs, as we described in our manuscript Introduction: “Lack of adequate ANC, household air pollution from unclean cooking fuels, open defecation, no access to improved water, violence, and other socioeconomic disparities are also considered important risk factors for ABO in sub-Saharan African (SSA) countries [15-17].”

However, this factor of “household air pollution from unclean cooking fuels” was not included in the study, as most women (∼80%) in the country use charcoal/coal as a cooking fuel, making comparison not feasible. 

The SES variable analysis is depicted in the Results section and disclosed in the Discussion: 

“Studies differ in their observations on risk, and contrary to expectations, no association was found with the sociodemographic maternal characteristics, and there was no relationship between ABO and the maternal sanitation behavior and type of access to water, as reported in other LMICs studies [15, 17, 67].”

---

## [Decision Letter · Decision Letter 1]

24 May 2023

PONE-D-22-27043R1Adverse birth outcomes and associated factors among newborns delivered in Sao Tome & Principe: a case‒control studyPLOS ONE

Dear Dr. Vasconcelos,

Thank you for submitting your manuscript to PLOS ONE. After careful consideration, we feel that it has merit but does not fully meet PLOS ONE’s publication criteria as it currently stands. Therefore, we invite you to submit a revised version of the manuscript that addresses the points raised during the review process.

The second reviewer has suggested important areas for further research. This points to the limitations of the current study in terms of the range of risk factors examined; this should be captured briefly in the limitations section.   

We look forward to receiving your revised manuscript.

Kind regards,

Calistus Wilunda, DrPH

Academic Editor

PLOS ONE

Journal Requirements:

Additional Editor Comments (if provided):

In the tables, please report odds ratios and 95% CIs to two decimal places.

In the abstract and results, it is incorrect to say that fewer than 8 ANC visits was protective of ABO. It should be 8 or more ANC visits, as shown in Table 3. You have also stated in the Discussion that “…complete ANC with eight or more contacts was a protective factor...”

Reviewers' comments:

Reviewer's Responses to Questions

**Comments to the Author**

1. If the authors have adequately addressed your comments raised in a previous round of review and you feel that this manuscript is now acceptable for publication, you may indicate that here to bypass the “Comments to the Author” section, enter your conflict of interest statement in the “Confidential to Editor” section, and submit your "Accept" recommendation.

Reviewer #1: All comments have been addressed

Reviewer #2: All comments have been addressed

2. Is the manuscript technically sound, and do the data support the conclusions?

Reviewer #1: Partly

Reviewer #2: Partly

3. Has the statistical analysis been performed appropriately and rigorously? 

Reviewer #1: I Don't Know

Reviewer #2: Yes

4. Have the authors made all data underlying the findings in their manuscript fully available?

Reviewer #1: No

Reviewer #2: Yes

5. Is the manuscript presented in an intelligible fashion and written in standard English?

Reviewer #1: Yes

Reviewer #2: Yes

6. Review Comments to the Author

Reviewer #1: Thanks, authors, for your efforts in refining the paper further. More work needs to be done on how you present the paper’s paragraphing, and readability. Please can you improve on this based on my comment below:

INTRODUCTION

1. Referring to the “introduction,” I suggest you remove the point and interval estimates mentioned as “ (aOR 2.6, CI: 1.4–4.8)”

2. Referring to the sentence ….“A study on maternal health during pregnancy found that women who had at least one health problem during their pregnancy had a twofold higher risk of delivering LBW newborns than women without any health problems (aOR 2.6, CI: 1.4–4.8) [14]….” Can you be specific on what health problems? Are these health problems similar to the ones mentioned in the next sentences?

3. Can the paragraph which followed “Most risk factors contributing to ABOs….” be tied to the next one or embedded into the paragraph which reads “ Sao Tome & Principe (STP) is an LMIC SSA country, with limited data on the overall ABO rate at the country level, and in the current era of the Sustainable Deve….” This paragraph seemed isolated. A paragraph should ideally have a topic sentence that focuses on a certain theme. I think you are trying to provide the uniqueness of different contexts and how ABOs and exposure are diversely occurring in other contexts, and interventions are well suited. I think it can fit into the next paragraph well.

4. Please, can you rewrite this paragraph? ….. This present study is included in a broader project on neonatal health in STP [24-28], and the authors studied the determinants for perinatal and neonatal mortality in another study. This current study aimed to identify the factors associated with ABOs among newborns delivered at the only hospital maternity unit in this country.”….. It does not make sense to me. You can also tie this paragraph to the next one under one theme.

MATERIALS AND METHODS

1. Please organize your paragraphs well. Under the section “ Selection of cases and controls,” the whole chunk should be a paragraph on its own.

2. Under the section “Sample size determination and sampling procedures,” Can you improve the readability of this sentence “Consenting participants in the sample were interviewed only after delivery and were followed-up (mother and newborn dyads) throughout their stays until hospital discharge.” I think you should use the word consented and not consenting.

3. Referring to the subheading “Operational definition of variables.” I think this chunk of information should be presented in the introduction, where you set the background of the study.

DISCUSSION/CONCLUSION

1. Lines 395, 399. These sentences cannot be paragraphs on their own. Can you fix these paragraphs ?

2. Please fix the paragraph under the strength and limitation and discussion section

Reviewer #2: Thank you authors for responding to R1. However, when considering adverse birth outcomes and associated factors in low- and middle-income countries (LMICs), several new research areas should be considered. These include, but not limited to:

1. Maternal Health Conditions: Exploring specific maternal health conditions and their association with adverse birth outcomes is important. Research should focus on conditions such as maternal infections (e.g., HIV, malaria), chronic diseases (e.g., diabetes, hypertension), mental health disorders, and nutritional deficiencies. Understanding the impact of these conditions on birth outcomes can guide prevention, management, and treatment strategies.

2. Healthcare Access and Quality: Assessing healthcare access and quality is crucial in LMICs. Research should examine the availability, affordability, and utilization of maternal healthcare services, including antenatal care, skilled birth attendance, emergency obstetric care, and postnatal care. Investigating the association between healthcare access, quality, and adverse birth outcomes can inform improvements in healthcare delivery.

3. Reproductive and Obstetric Interventions: Evaluating the impact of specific reproductive and obstetric interventions on adverse birth outcomes is necessary. Research should focus on interventions such as antenatal corticosteroids, magnesium sulfate for pre-eclampsia, and timely cesarean section. Assessing the effectiveness and implementation of these interventions in LMICs can guide evidence-based practices.

4. Environmental Factors: Understanding the role of environmental factors in adverse birth outcomes is important. Research should explore the impact of air pollution, water contamination, exposure to toxins, and indoor biomass fuel use on birth outcomes. Identifying environmental risk factors and their association with adverse outcomes can inform policies and interventions to mitigate exposures.

5. Infectious Diseases: Investigating the impact of infectious diseases on adverse birth outcomes is critical. Research should explore the association between diseases such as Zika, malaria, syphilis, and adverse outcomes such as preterm birth, low birth weight, and congenital anomalies. Understanding the mechanisms and developing effective preventive and treatment strategies is essential.

6. Intersectionality: Recognizing the intersectionality of factors influencing adverse birth outcomes is crucial. Identifying vulnerable subgroups and understanding their unique challenges can guide targeted interventions.

7. Long-Term Outcomes: Investigating the long-term consequences of adverse birth outcomes is necessary. Research should assess the impact of adverse outcomes on child development, growth, and health outcomes in later life. Understanding the long-term implications can guide early interventions and support systems for affected children and families.

Though this case-control study is limited by scope, etc., considering and highlighting these new research areas (backed by latest literature) on adverse birth outcomes and associated factors in LMICs (in Discussion section) will deepen our understanding of the complex determinants and inform future R&D in evidence-based interventions to improve birth outcomes and maternal and child health.

7. PLOS authors have the option to publish the peer review history of their article (what does this mean?). If published, this will include your full peer review and any attached files.

Reviewer #1: **Yes: **Lydia Kaforau

Reviewer #2: **Yes: **Dr Amanuel Abajobir

---

## [Author Response · Author response to Decision Letter 1]

9 Jun 2023

Rebuttal letter

Point-by-point response to Reviewers

R2 for manuscript PONE-D-22-27043R1

Adverse birth outcomes and associated factors among newborns delivered in Sao Tome & Principe: a case‒control study

PLOS ONE

Dear Editor and Reviewers, 

We truly appreciate the Academic Editor Calistus Wilunda DrPH and Reviewer´s Lydia Sandrah Kaforau and Dr Amanuel Abajobir feedback and precious contributions for this R2.

We now submit a R2 version of the manuscript that addresses the points raised during the review process. In the following detailed response, we address each comment calling for changes point-by-point, indicating where relevant additional texts have been added to the body of the manuscript. 

We submit a manuscript with track changes (file labeled 'Revised Manuscript with Track Changes') and a clean version of the review manuscript (“Manuscript”). 

We believe that with the changes made, the manuscript has improved substantively in view of being considered for publication.

PONE-D-22-27043R1

Adverse birth outcomes and associated factors among newborns delivered in Sao Tome & Principe: a case‒control study

PLOS ONE

Dear Dr. Vasconcelos,

Thank you for submitting your manuscript to PLOS ONE. After careful consideration, we feel that it has merit but does not fully meet PLOS ONE’s publication criteria as it currently stands. Therefore, we invite you to submit a revised version of the manuscript that addresses the points raised during the review process.

The second reviewer has suggested important areas for further research. This points to the limitations of the current study in terms of the range of risk factors examined; this should be captured briefly in the limitations section. 

We look forward to receiving your revised manuscript.

Kind regards,

Calistus Wilunda, DrPH

Academic Editor

PLOS ONE

Authors’ response: 

Dear Dr. Calistus Wilunda PhD, we truly appreciate your valuable feedback and suggestions. 

We now submit a revised manuscript adding the topic “Future research” and amending the “Strengths and Limitations” section as by your suggestion. 

References were amended accordingly to corrections and three additional references were included taking into consideration Reviewer 2´s concerns. 

We believe that with the changes made, the manuscript R2 has improved substantively in view of being considered for publication.

Journal Requirements:

Additional Editor Comments (if provided):

In the tables, please report odds ratios and 95% CIs to two decimal places.

Authors’ response: 

Done. 

In the abstract and results, it is incorrect to say that fewer than 8 ANC visits was protective of ABO. It should be 8 or more ANC visits, as shown in Table 3. You have also stated in the Discussion that “…complete ANC with eight or more contacts was a protective factor...”

Authors’ response: 

Done. The manuscript (abstract, discussion and conclusions) was amended as suggested.

Reviewers' comments:

Reviewer's Responses to Questions

Comments to the Author

1. If the authors have adequately addressed your comments raised in a previous round of review and you feel that this manuscript is now acceptable for publication, you may indicate that here to bypass the “Comments to the Author” section, enter your conflict of interest statement in the “Confidential to Editor” section, and submit your "Accept" recommendation.

Reviewer #1: All comments have been addressed

Reviewer #2: All comments have been addressed

2. Is the manuscript technically sound, and do the data support the conclusions?

Reviewer #1: Partly

Reviewer #2: Partly

3. Has the statistical analysis been performed appropriately and rigorously?

Reviewer #1: I Don't Know

Reviewer #2: Yes

4. Have the authors made all data underlying the findings in their manuscript fully available?

Reviewer #1: No

Reviewer #2: Yes

Authors’ response: 

The datasets used and analyzed during the current study are attached as “supporting information”.

5. Is the manuscript presented in an intelligible fashion and written in standard English?

Reviewer #1: Yes

Reviewer #2: Yes

6. Review Comments to the Author

Reviewer #1: 

Thanks, authors, for your efforts in refining the paper further. More work needs to be done on how you present the paper’s paragraphing, and readability. 

Authors’ response: 

We truly appreciate Reviewer 1´s valuable contributions and feedback. We now submit a revised manuscript with the improvements suggested for this R2.

Please can you improve on this based on my comment below:

INTRODUCTION

1. Referring to the “introduction,” I suggest you remove the point and interval estimates mentioned as “ (aOR 2.6, CI: 1.4–4.8)”

Authors’ response: 

Done. 

2. Referring to the sentence ….“A study on maternal health during pregnancy found that women who had at least one health problem during their pregnancy had a twofold higher risk of delivering LBW newborns than women without any health problems (aOR 2.6, CI: 1.4–4.8) [14]….” Can you be specific on what health problems? Are these health problems similar to the ones mentioned in the next sentences?

Authors’ response: 

The comment is quite pertinent; however, the authors of the study [ref 14] did not provide the description of which maternal co-morbidities were included. Only the general term “co-morbidity” (yes or no) was considered. 

This article was published in a 2020 PLOs One (KC A, Basel PL, Singh S. Low birth weight and its associated risk factors: Health facility-based case-control study. https://doi.org/10.1371/journal.pone.0234907).

Nonetheless, having a co-morbidity or a health problem is an important, well-established, maternal risk factor for ABOs, and thus, we think that this information should be included in our manuscript.

Therefore, we amended the two sentences, as follows:

“A study on maternal health during pregnancy found that women who had at least one co-morbidity during their pregnancy had a twofold higher risk of delivering LBW babies than women without any health problems [14]. Indeed, studies report that antepartum infections such as malaria, syphilis and others, and noninfectious conditions such as anemia, hypertension, hyperglycemia, and obstetric complications are all linked to ABOs [9-12].

3. Can the paragraph which followed “Most risk factors contributing to ABOs….” be tied to the next one or embedded into the paragraph which reads “Sao Tome & Principe (STP) is an LMIC SSA country, with limited data on the overall ABO rate at the country level, and in the current era of the Sustainable Deve….” This paragraph seemed isolated. A paragraph should ideally have a topic sentence that focuses on a certain theme. I think you are trying to provide the uniqueness of different contexts and how ABOs and exposure are diversely occurring in other contexts, and interventions are well suited. I think it can fit into the next paragraph well.

Authors’ response: 

We agree with Reviewers 1´s suggestion. The paragraph was amended as suggested. 

4. Please, can you rewrite this paragraph? ….. This present study is included in a broader project on neonatal health in STP [24-28], and the authors studied the determinants for perinatal and neonatal mortality in another study. This current study aimed to identify the factors associated with ABOs among newborns delivered at the only hospital maternity unit in this country.”….. It does not make sense to me. You can also tie this paragraph to the next one under one theme.

Authors’ response: 

We agree with Reviewers 1´s suggestion. The paragraph was amended as suggested. 

MATERIALS AND METHODS

1. Please organize your paragraphs well. Under the section “Selection of cases and controls,” the whole chunk should be a paragraph on its own.

Authors’ response: 

The paragraph regarding “Selection of cases and controls was amended as suggested. The variable definition for ABOs were moved for the topic “Operational definition of variables.” 

2. Under the section “Sample size determination and sampling procedures,” Can you improve the readability of this sentence “Consenting participants in the sample were interviewed only after delivery and were followed-up (mother and newborn dyads) throughout their stays until hospital discharge.” I think you should use the word consented and not consenting.

Authors’ response: 

We agree with Reviewers 1´s suggestion.

3. Referring to the subheading “Operational definition of variables.” I think this chunk of information should be presented in the introduction, where you set the background of the study.

Authors’ response: 

We can follow the Reviewers 1 rationale, but we were asked to fulfill PLOS One guidelines and use STROBE statement and requirements of the STROBE guidelines for reporting an observational study – i.e. cohort, case-control, and cross-sectional studies.

Therefore, to comply with the above-mentioned recommendations the operational definition of variables should be described in the Materials and Methods section. Moreover, we also believe that this section concerning the “Operational definition of variables” will benefit the readers, as it explains and clarifies how the most important variables were operationalized in this study. We deleted those that did not need to be further specified. 

DISCUSSION/CONCLUSION

1. Lines 395, 399. These sentences cannot be paragraphs on their own. Can you fix these paragraphs ?

Authors’ response: 

We agree with Reviewers 1´s suggestion. The paragraph was amended as suggested. 

2. Please fix the paragraph under the strength and limitation and discussion section

Authors’ response: 

This section was amended as suggested by both reviewers. 

Reviewer #2: 

Thank you authors for responding to R1. 

Authors’ response: 

We truly appreciate Reviewer 2´s valuable contributions and feedback. We now submit a revised manuscript with improvements as suggested for this R2.

However, when considering adverse birth outcomes and associated factors in low- and middle-income countries (LMICs), several new research areas should be considered. 

These include, but not limited to:

1. Maternal Health Conditions: Exploring specific maternal health conditions and their association with adverse birth outcomes is important. Research should focus on conditions such as maternal infections (e.g., HIV, malaria), chronic diseases (e.g., diabetes, hypertension), mental health disorders, and nutritional deficiencies. Understanding the impact of these conditions on birth outcomes can guide prevention, management, and treatment strategies.

Authors’ response:

The main maternal health conditions frequently reported to be associated with ABOs were included in this study, namely, infectious conditions (malaria, HIV, syphilis, hepatitis B, and bacteriuria) and non-infectious conditions (maternal anemia, hyperglycemia, twin pregnancy) and maternal characteristics (sociodemographic features, preconception factors, ANC contacts and screenings). 

As previously disclosed, there are some limitations regarding the maternal factors such as gestational weight gain, BMI, and height that were not analyzed in this study, as these measurements are not done during ANC contacts in the country, missing key mediators for ABOs. This information in included in the “Strength and Limitations” section.

2. Healthcare Access and Quality: Assessing healthcare access and quality is crucial in LMICs. 

Research should examine the availability, affordability, and utilization of maternal healthcare services, including antenatal care, skilled birth attendance, emergency obstetric care, and postnatal care. Investigating the association between healthcare access, quality, and adverse birth outcomes can inform improvements in healthcare delivery.

Authors’ response:

Healthcare access and quality were analyzed in this study, using the following variables i) antenatal care (partial ANC <4, adequate with ANC 4-8 and complete with ANC 8 or +, timing of first ANC booking and having done an obstetric ultrasound), ii) skilled birth attendance (obstetrician and midwife), iii) emergency obstetric care (pregnant women needed to be transferred from another unit and fetal distress at birth), and iv) postnatal care (neonatal resuscitation performed, admission at NCU and antibiotic administration). 

3. Reproductive and Obstetric Interventions: Evaluating the impact of specific reproductive and obstetric interventions on adverse birth outcomes is necessary. Research should focus on interventions such as antenatal corticosteroids, magnesium sulfate for pre-eclampsia, and timely cesarean section. Assessing the effectiveness and implementation of these interventions in LMICs can guide evidence-based practices.

Authors’ response: 

Pre-ecamplsia, cesarean section, instrumental vaginal delivery and other major intrapartum factors as obstructed labor, postpartum hemorrhage, meconium and PROM were analyzed in this study. 

Other interventions, such as antenatal corticosteroids are mainly aimed at evaluating preterm survival/morbidity as well as magnesium sulfate. In this study, they were not included since our main goal was to identify overall ABOs, disregarding survival or morbidity outcomes.

The main reproductive and preconception factors reported to be associated with ABOs were also included in this study, namely, type of pregnancy plan, parity, previous abortion, previous stillbirth, poor birth spacing, and previous cesarean section. 

4. Environmental Factors: Understanding the role of environmental factors in adverse birth outcomes is important. Research should explore the impact of air pollution, water contamination, exposure to toxins, and indoor biomass fuel use on birth outcomes. Identifying environmental risk factors and their association with adverse outcomes can inform policies and interventions to mitigate exposures.

Authors’ response: 

The environmental factors included in this study were access or not to improved water and the type of sanitation practice. In STP, the kitchen is outside the house, but women still use unclean fuel for cooking - coal, charcoal, wood, kerosene - all known to be associated with ABOs, but practiced by almost all participants, thus, no different results between cases and controls would be expected. 

Air pollution, water contamination, exposure to toxins, and indoor biomass fuel use, fortunately, are not a current problem in this country, although we are aware that these factors are known to be a huge concern in other SSA countries. 

Sao Tome & Principe is known for its rich biodiversity and natural landscapes, home to lush rainforests, endemic plant and animal species, and known for its diverse ecosystems. In contrast, STP is still a very significant healthcare resource-constrained country with a substantial adverse health impact on the population. 

5. Infectious Diseases: Investigating the impact of infectious diseases on adverse birth outcomes is critical. Research should explore the association between diseases such as Zika, malaria, syphilis, and adverse outcomes such as preterm birth, low birth weight, and congenital anomalies. Understanding the mechanisms and developing effective preventive and treatment strategies is essential.

Authors’ response: 

We appreciate Reviewer 2´s comment. 

We included a new paragraph that clarifies the infectious disease epidemiology in this setting that supports the lack of association found in this case-control study:

“In this study, no association was found between maternal infectious diseases (malaria, HIV, syphilis) and ABOs. This is mainly due to the fact that STP, in contrast to most SSA countries, is about to reach the elimination of HIV, syphilis, hepatitis B and malaria supporting the reduced number of mothers infected enrolled in this study [68,69]. Other viruses known to cause ABOs in SSA countries, such as Zika and dengue were not evaluated in this study since there is no evidence of Zika virus transmission in the country, and the first dengue fever outbreak in STP occurred two years after the completion of the field study [70].” 

6. Intersectionality: Recognizing the intersectionality of factors influencing adverse birth outcomes is crucial. Identifying vulnerable subgroups and understanding their unique challenges can guide targeted interventions.

Authors’ response: 

Throughout this study we intended to identify vulnerable subgroups that are well known to be at a higher risk for ABOs, namely the maternal subgroups with extreme age groups, adolescents, and elderly mothers. Therefore, we stratified maternal age as 14-16 years old (early adolescents), 17-19 (late adolescent), 20-34 and >35 years (elderly pregnant) but we did not find a significant association with ABOs in this study. 

Moreover, other vulnerable subgroups, such as, single mothers, unemployed, illiterate or with low education level and those living in rural areas were also not associated as having higher risks for ABOs in this study. 

7. Long-Term Outcomes: Investigating the long-term consequences of adverse birth outcomes is necessary. Research should assess the impact of adverse outcomes on child development, growth, and health outcomes in later life. Understanding the long-term implications can guide early interventions and support systems for affected children and families.

Authors’ response: 

We completely agree with Reviewer 2 statement. Future studies should be conducted for assessing the impact of ABOs on child development, growth, and health outcomes in later life in this resource-constrained setting. 

Though this case-control study is limited by scope, etc., considering and highlighting these new research areas (backed by latest literature) on adverse birth outcomes and associated factors in LMICs (in Discussion section) will deepen our understanding of the complex determinants and inform future R&D in evidence-based interventions to improve birth outcomes and maternal and child health.

Authors’ response: 

Considering all the abovementioned factors, we consider that the most important and frequent factors associated with ABOs in LMICs, according to the recent literature, were analyzed in this case-control study. 

To better address Reviewer 2´s concerns and following the Edito´s suggestions regarding this subject, we amended the section “Strengths and limitations” and included a new topic “Future research”, as follows: 

“Strengths and limitations

In this study, all clinical pregnancy-related information was abstracted from ANC cards and maternity registers to limit recall bias. Moreover, this study analyzed the main variables that are most frequently associated with ABOs in LMICs, including factors related to i) maternal health conditions, ii) infectious diseases, iii) environmental factors, iv) healthcare access, and v) obstetric interventions. 

Regarding the limitations, maternal factors such as gestational weight gain, BMI, and height were not analyzed in this study, as these measurements are not performed during ANC contacts in the country, missing key mediators for ABOs [12]. Participants were asked about the weight gained during the pregnancy or their height, but mothers were not aware of either their height or weight gain. This lack of knowledge of basic one´s features and body conscience might also be due to the lack of scales and stadiometers in most healthcare facilities.

Notwithstanding these limitations, this study represents a significant first step in the description of ABOs and key modifiable associated factors in this resource-constrained setting.

Future research

The scope of this case-control study missed-out some recent areas of research, such as the impact of water contamination, traditional herbs, exposure to toxins on birth outcomes and long-term implications of ABOs on child development, growth, and health outcomes, which should be a matter of future research in this country [15-17]. This will promote further understanding of the complex determinants of ABOs and provide new clues on how to improve birth outcomes and maternal and child health.

7. PLOS authors have the option to publish the peer review history of their article (what does this mean?). If published, this will include your full peer review and any attached files.

Do you want your identity to be public for this peer review? For information about this choice, including consent withdrawal, please see our Privacy Policy.

Reviewer #1: Yes: Lydia Kaforau

Reviewer #2: Yes: Dr Amanuel Abajobir

---

## [Editor Report · Decision Letter 2]

19 Jun 2023

Adverse birth outcomes and associated factors among newborns delivered in Sao Tome & Principe: a case‒control study

PONE-D-22-27043R2

Dear Dr. Vasconcelos,

We’re pleased to inform you that your manuscript has been judged scientifically suitable for publication and will be formally accepted for publication once it meets all outstanding technical requirements.

Kind regards,

Calistus Wilunda, DrPH

Academic Editor

PLOS ONE

Additional Editor Comments (optional):

Carefully proofread the manuscript for typographical errors before publication. I have noticed one error: "SSA African country". "African" should be deleted.
---

## [Editor Report · Acceptance letter]

27 Jun 2023

PONE-D-22-27043R2 

Adverse birth outcomes and associated factors among newborns delivered in Sao Tome & Principe: a case‒control study 

Dear Dr. Vasconcelos:

I'm pleased to inform you that your manuscript has been deemed suitable for publication in PLOS ONE. Congratulations! Your manuscript is now with our production department. 

Kind regards, 

on behalf of

Dr. Calistus Wilunda 

Academic Editor

PLOS ONE